# A Study on the Simulation and Experiment of Evaporative Condensers in an R744 Air Conditioning System

**DOI:** 10.3390/mi14101826

**Published:** 2023-09-25

**Authors:** Thanhtrung Dang, Hoangtuan Nguyen

**Affiliations:** Department of Thermal Engineering, HCMC University of Technology and Education, Ho Chi Minh City 71307, Vietnam; tuannh@hcmute.edu.vn

**Keywords:** evaporative condenser, heat transfer, R744, subcritical, air conditioning

## Abstract

The heat transfer characteristics of evaporative condensers in an R744 air conditioning system were evaluated using the numerical and the experimental methods. Two configurations of condensers were studied: Case 1 with five layers of tubes and Case 2 with eight layers of tubes. In order to evaluate the heat transfer characteristics, the temperature field, the phase change, the pressure distribution, and thermodynamic parameters were considered. For Case 2, it indicated the capability of R744 condensation from the superheated status to the liquified status by analyzing the outlet temperature of the condenser changed from 28.7 °C to 30.3 °C with a change in condensation pressure from 72.6 bar to 68.5 bar. In this study, R744 mass flow rate increases from 14.34 kg/h to 46.08 kg/h, and the pressure drop also increases from 0.23 bar to 0.47 bar for the simulation and 0.4 bar to 0.5 bar for the experiment, respectively. The results indicate that the five-layer configuration causes a higher pressure drop and lower COP than those obtained from the eight-layer one (splitting into two sets for smaller pressure drop). Furthermore, the evaporative condensers using mini tubes that are flooded in the cooling water tank are suitable for the subcritical R744 air conditioning system. In addition, the results obtained from the experimental data are in good agreement with those obtained from the numerical results, with a deviation of less than 5%.

## 1. Introduction

Currently, scientists are concerned about greenhouse effect problems. One of these problems is the exhaust gas that is generated by refrigerant leakage from air conditioning systems as well as industrial and commercial refrigeration systems. Therefore, research on environmentally friendly refrigerants is an interesting goal for experts. In natural refrigerants, R744 (known as CO_2_) has the potential to replace traditional refrigerants with many advantages such as having low Global Warming Potential (GWP = 1) and Ozone Depletion Potential (ODP = 0) and not being able to cause fire or explosion. Based on the above advantages, many previous studies have been conducted to experimentally evaluate subcritical and transcritical CO_2_ refrigeration systems. 

The R744 transcritical cycle has been widely researched to find optimal working points as well as optimal designs to improve compressor energy efficiency. Related to this field, Zhang et al. [1] conducted simulations based on the R744 transcritical cycle involving a few performance parameters and CFD (Computational Fluid Mechanics) models for tubular and wing air cooling condensers. The models were compared and confirmed with experimental measurements and literature reviews. The simulation and experiment results in [2,3,4] were analyzed with an R744 transcritical cycle operating in heat pump mode with auxiliary thermoelectric cooling equipment. These studies show the effects of coolant flow rate and temperature, auxiliary cooling capacity, and compressor discharge pressure on the COP (Coefficient of Performance) and the heating efficiency factor. Deng et al. [5] theoretically analyzed an ejector refrigeration cycle operating on a transcritical cycle. Using an expansion device and controlling the suction rate, the ejector ensures the amount of vapor returning to the compressor and the amount of liquid to the evaporator. The COP was studied in a variety of injection ratios while simultaneously comparing it with conventional vapor compression refrigeration cycles. Zhang et al. [6,7] experimentally compared the R744/R290 subcritical cycle in the cascade system to the R744 transcritical cycle. The analysis results of the discharge pressure, the outlet temperature of the gas cooler, and the mass ratio between R744 and R290 show that using an IHX (Internal Heat Exchanger) for the subcritical cycle is less effective than the transcritical cycle. Gullo et al. [8] evaluated aspects related to the R744 transcritical cycle in a refrigeration plant in supermarkets. The study concluded that R744 can be favorably utilized at supermarkets, especially suitable for areas with high ambient temperatures. Hazarika et al. [9] conducted an experimental study and numerical simulation of an R744 transcritical air conditioning system. Not only were the variations in the heat transfer area of the evaporator proposed, to study the system COP further, but also an evaluation of the effect of air velocity and pressure of the air cooler was presented. Khedher et al. [10] predicted the heat loss of buildings by applying artificial neural networks to study the walls of buildings and coating materials with respect to the effect of the heat transfer coefficient as well as indoor and outdoor surface temperatures. However, the studies in [1,2,3,4,5,6,7,8,9,10] did not mention CO_2_ air conditioning systems with throttle inlet temperatures below the critical point.

To reduce the condensation pressure and increase the overall refrigeration performance, Sanz-Kock et al. [11] conducted an experimental evaluation of the energy efficiency of a cascade refrigeration system using R134a/CO_2_, where they used semi-hermetic compressors in subcritical cycles. Parameters such as compressor efficiency, temperature difference at the cascade heat exchanger, and compressor discharge temperature were evaluated under conditions of various evaporation and condensation temperatures. Zhang et al. [12] proposed a cascade refrigeration system with R1270/CO_2_. Particularly, R1270 was used in the high temperature cycle, while CO_2_ was used as a refrigerant in the low temperature cycle. The results showed that R1270 is a better replacement than R290 and R717 in the high-stage cycle. Llopis et al. [13] implemented an experimental evaluation of welded plate heat exchangers in subcritical cycles using R744 with water cooling evaporative condenser and semi-hermetic compressors. The influence of the internal heat exchanger in the cascade cycle was analyzed theoretically. The results indicate that welded plate heat exchangers for subcritical cycles had less impact on energy efficiency but were suitable for energy improvement in the cascade cycles. Nicola et al. [14] analyzed the thermal efficiency of the cascade refrigeration cycle, in which the low-stage R744 substance was mixed with other refrigerants such as R170, R290, R1150, and R717. Based on the Carnahane–Starlinge–DeSantis equation of state, the COP results indicated that R744 should only be used in its mono-component form, in combination with other refrigerants. The experimental results in [15,16] discussed the operating parameters and energy efficiency in a cascade cycle using the R717/R744. Dopazo et al. [15] performed a comparison with a two-stage cycle using R717 under the same operating conditions with evaporation temperatures and condensation temperatures corresponding to cascade cycles ranging from −17.5 °C to −7.5 °C. Lee et al. [16] conducted a further exploration of the evaporation temperature, the condensation temperature, and the temperature difference in the cascade heat exchanger. From refs. [11,12,13,14,15,16], most of the research conducted on subcritical cycles focused on experimental research to find suitable refrigerants in the high-temperature cycle, find out the relationship between COP and the operation parameters, and determine the optimal operation regime. They used the cascade refrigeration system and did not use the single-stage subcritical refrigeration system.

From the literatures above, the studies have focused on exploring the influence of parameters such as evaporation temperature, condensation temperature, condensation pressure, different ways of configuring equipment, design parameters, and refrigerants to find the most suitable case in each specific application. While there are fewer R744 subcritical cycle applications that use evaporative condensers using water and air, especially with the condensing mini tubes that are flooded in the cooling water tank. Therefore, the numerical simulation and experimental investigation for this application are potential directions that should be developed, especially in tropical countries. To sum up, the numerical simulation on and experimental verification of the heat transfer characteristics of flooded-type condenser applied in residential air conditioning systems is one of the breakthrough studies that will be presented in more detail in this paper. 

## 2. Methodology

### 2.1. Calculation Design

With the initial design data for an R744 air conditioning system, the testing cycle has a cooling capacity of 1000 W, an evaporation temperature of 10 °C, and a condensation temperature of 30.5 °C. Design parameters were calculated and expressed according to Figure 1 and Table 1.

In this study, the key components of a testing cycle were designed based on the computational theory with the following equations:

The cooling capacity was calculated as follows:(1)$$Q_{0}=G\left(h_{5}-h_{4}\right)$$

The condensing capacity for G kg was calculated as follows:(2)$$Q_{k}=G\left(h_{2}-h_{3^{\prime }}\right)$$

The work of adiabatic compression was determined as follows:(3)$$N=G\left(h_{2}-h_{1}\right)$$

The Coefficient of Performance (COP) of the cycle was calculated as follows:(4)$$\mathrm{COP}=\frac{Q_{0}}{N}$$

The overall heat transfer rate was calculated as follows:(5)$$A=\frac{Q_{k}}{U\Delta t_{\mathrm{lm}}}$$
where $Q_{o}$ is cooling capacity (W); G is mass flow rate (kg^−1^); h is enthalpy of fluid (kJ·kg^−1^); $Q_{k}$ is condensing capacity (W); N is adiabatic compression power (kW); A is the heat transfer area (m^2^); U is the overall heat transfer coefficient (W·m^−2^·K^−1^); and $\Delta t_{\mathrm{lm}}$ is log mean temperature difference (°C). 

Based on the same overall heat transfer coefficient, this study offers two design configurations for the condenser, consisting of Case 1 (five layers of copper tubes) and Case 2 (eight layers of copper tubes), with dimensions and parameters shown in Figure 2 and Table 2. These condensers use mini-size tubes. For Case 2, there are two tube sets (banks) arranged in parallel; each set has four tube layers (or four passes). Each layer has 25 mini tubes with the outer and inner diameters of 4 mm and 3 mm, respectively. The actual image of the condenser with eight layers using mini tubes is shown in Figure 3. The condensing mini tubes are flooded in the cooling water tank.

### 2.2. Numerical Simulation

This model was solved based on Conjugate Heat Transfer model [17,18], which was formed of two models: (1) Heat Transfer in Solids and Fluids and (2) Turbulent Flow.

For Heat Transfer in Solids and Fluids model, the assuming equations are expressed as follows:(6)$${\rho C}_{p}u\nabla T+\nabla q=Q+Q_{\mathrm{ted}}$$
(7)q=−k∇T

For the Turbulent Flow model, the assuming equations are expressed as follows:(8)$$\rho \left(u\nabla \right)u=\nabla \left[-\mathrm{pI}+K\right]+F$$
(9)$$\nabla \left(\rho u\right)=0$$

The transport equation for k reads is as follows:(10)$$\rho \left(u\nabla \right)k=\left[\left(\mu +\frac{\mu_{T}}{\sigma_{k}}\right)\right]\nabla k+P_{k}-\rho \epsilon \;\mathrm{where}\;\;K=\left(\mu +\mu_{T}\right)\left(\nabla u+{\left(\nabla u\right)}^{T}\right)-\frac{2}{3}\left(\mu +\mu_{T}\right)\left(\nabla u\right)I-\frac{2}{3}\rho \mathrm{kI}$$

The transport equation for ϵ is as follows:(11)$$\rho \left(u\nabla \right)\epsilon =\left[\left(\mu +\frac{\mu_{T}}{\sigma_{\epsilon }}\right)\right]+C_{\epsilon 1}\frac{\epsilon }{k}P_{k}-C_{\epsilon 2}\rho \frac{\epsilon^{2}}{k},$$
(12)$$\mathrm{Where}\;\epsilon =\mathrm{ep}\;\mathrm{and}\;\mu_{T}=C_{\mu }\rho \frac{k^{2}}{\epsilon }$$

The momentum formulation is as follows:(13)$$P_{k}=\mu_{T}\left[\nabla u÷\left(\nabla u+{\left(\nabla u\right)}^{T}\right)-\frac{2}{3}{\left(\nabla u\right)}^{2}\right]-\frac{2}{3}\rho k\nabla u$$

The phase change equation is as follows:(14)$$\rho =\theta_{1}\rho_{1}+\theta_{2}\rho_{2}$$

The specific heat capacity at constant pressure is expressed as follows:(15)$$C_{p}=\frac{1}{\rho }\left(\theta_{1}\rho_{1}C_{p,1}+\theta_{2}\rho_{2}C_{p,2}\right)+L_{1-2}\frac{\partial \alpha_{m}}{\partial T}$$

The mass fraction of two-phase flow is as follows:(16)$$\alpha_{m}=\frac{1}{2}\frac{\theta_{2}\rho_{2}-\theta_{1}\rho_{1}}{\theta_{1}\rho_{1}+\theta_{2}\rho_{2}}$$

The relationship between the vapor quality and the phase indicator is as follows:(17)$$x=\theta_{2}\;\mathrm{and}\;1-x=\theta_{1}$$
where $\mu_{T}$ is the turbulence viscosity (Pa·s); u is velocity of fluid (m·s^−1^); ρ is density of fluid (kg·m^−3^); k is thermal conductivity (W·m^−1^·K^−1^); T is the absolute temperature (K), C_p_ is the specific heat capacity at constant pressure (J·kg^−1^·K^−1^)), Q contains additional heat sources (W·m^−3^), q is the external heat flux (W·m^−2^); θ is phase indicator fraction; α is thermal diffusion (W·m^−2^); ∇u is the gradient of velocity (m·s^−2^); and x is vapor quality.

The numerical simulation for water-cooled condensers was conducted using COMSOL Multiphysics 6.1 software with the Conjugate Heat Transfer model [17]. This model consists of two models: the Turbulent Flow model and Low-Reynolds k-ε combined with the Heat Transfer in Solids and Fluids model.

To simplify the simulation model for Case 2, there is an even division of rows of pipes between two sets; the R744 entering the condenser is evenly divided. Therefore, the simulation model uses a symmetrical method by reducing the number of elements, saving time, improving convergence, and increasing accuracy.

Complete mesh predefined as normal level consists of 692,399 domain elements, 134,042 boundary elements, and 24,441 edge elements as shown in Figure 4. With the quality of the simulation mesh supporting a faster simulation process, the average Skewness acceptable indexes in the rows of pipes at the condenser are from 0.5–0.7 for the tubes and from 0.75–0.8 for the manifolds, as shown in Figure 5. A grid independence analysis was implemented to verify the simulation, as shown in Table 3. Regardless of mesh quality, the R744 outlet temperature converges at 28.7 °C. 

For heat transfer variables, the GMRES (Generalized Minimal Residual Method) iterative solver was used. This method produces an approximate solution to the linear system after a finite number of steps. These methods are useful for large systems of equations where it is reasonable to trade-off precision for a shorter run time. Iterative methods use the coefficient matrix indirectly, through a matrix–vector product or an abstract linear operator. Iterative methods can be used with any matrix, but they are typically applied to large sparse matrices for which direct solves are slow. Furthermore, for turbulence variables, the PARDISO (Parallel Sparse Direct Solver) was used to improve sequential and parallel sparse numerical factorization performance by pipelining parallelism with a combination of left-looking and right-looking supernode techniques. PARDISO is a multithreaded on platform to save the solving time. In each case study, the boundaries conditions include the inlet temperature of R744 (°C), the outlet pressure of R744 (bar), the mass flow rate of R744 (g.s^−1^), the average water temperature (°C), and velocity of water (m/s).

The computer for solving this model has the following configuration: CPU-Intel Core i7 6820HQ (2.7 GHz Turbo 3.6 GHz, 4 cores, 8 threads), Ram of 16 GB DDR4 with bus 2400 MHz, and VGA-Nvidia Quadro M3000M. The average computation time fluctuates from 30 min to 2 h.

### 2.3. Experiment Set Up

The diagram for the testing system and the data acquisition establishment are shown in Figure 6. The experimental system is designed with a complete R744 air conditioning cycle (which is cooled by the evaporative cooling method) and operated in Vietnamese climatic conditions. For this experimental system (in Figure 6a), the low-pressure superheated vapor coming from the evaporator enters the compressor to perform the process of isotropy adiabatic compression that converts external work into high-pressure superheated vapor. The superheated vapor continues to enter the condenser, releasing heat to water, performing complete liquefied isobaric condensation. The liquid refrigerant proceeds to the hand throttle valve to become a wet saturated vapor with low pressure (low temperature). Moreover, R744 enters the evaporator-receiving heat from the ambient that performs isobaric evaporation to become the dry saturated vapor. Leaving the evaporator, R744 returns to the compressor to complete a closed cycle. A real photo of the system is shown in Figure 6b.

Furthermore, measuring devices such as flowmeters, temperature, and pressure at the cycle’s state points were installed at specified locations to collect the experimental parameters. The types of testing apparatuses and their accuracies are shown in Table 4. The experimental uncertainties were estimated following the method described by Holman [19]. The uncertainty values of several parameters such as the outlet temperature of condenser, the mass flow rate, and the pressure drop are 0.5%, 0.5%, and 0.7%, respectively.

The test loop operated for 20 min in order to have the system reach its steady state. Then, the experimental data (temperature, pressure, flow rate, etc.) were recorded for 10 min. The values for the system as a whole were calculated from the average values of all temperatures being recorded. 

## 3. Results and Discussion

### 3.1. Temperature Field at R744 Side in Numerical Simulation

The simulation model is solved with the boundary conditions, as shown in Table 5. For example, the inlet temperature of the condenser on the R744 side is 63.3 °C, and the mass flow rate of R744 is 3.28 g/s. These values were verified with the experimental data. 

Figure 7 shows the temperature field of R744 in the condenser for Case 2. Based on 3D images and temperature scales (Celsius), under current simulation conditions, the system operates under a pressure of 72.6 bar, and the superheated vapor enters the condenser at a temperature of 63.3 °C. It is observed that the temperature distribution through the unit has changed with the inlet R744 temperature of 63.3 °C after exchanging heat with water at the temperature of 26.4 °C (the average velocity of water is 0.003 m/s). The temperature field of the R744 at the inlet decreases gradually through each layer until the end of the condenser with the outlet temperature of 28.7 °C. Furthermore, Figure 8 indicates the phase change of R744, and the vapor phase is indicated as transparent, while the liquid phase is indicated on a blue scale. From the numerical simulation, it is determined that R744 is completely liquefied in the last two layers. This is the peculiarity that numerical simulation can show while the experiment is difficult to observe.

### 3.2. Validation

The outlet R744 temperatures of the condensers from the simulation results were compared experimentally to verify the reliability under the same conditions. Table 6 presents comparative data between simulation and experiment under typical operating conditions for R744 side: the evaporation temperature of 9.2 °C, the mass flow rate of 14.34 kg/h, the inlet pressure of 72.6 bar, and the inlet temperature of 63.3 °C. It is observed that the deviation number is quite small (1.9% with temperature and 0.3% with pressure), which demonstrates the predictive simulation is very good and reliable when compared to the experiment for studying the evaporative condenser.

### 3.3. The Effect of the Inlet and Outlet R744 Temperatures of Condenser

In Case 2, the evaporation temperature was varied by adjusting the opening of the throttle valve, which gradually increased the evaporation pressure in the range from 43.38 bar to 50 bar as well as the evaporation temperature in the range from 9.2 °C to 16.2 °C. In this case, the mass flow rates of R744 through the condenser are increasing from 14.34 kg/h to 46.08 kg/h. The boundary conditions for the simulation are shown in Table 7.

The data obtained from the experiments and the predictions from the simulation are shown in Figure 9 for Case 2. In the study cases, the outlet R744 temperatures of the condenser gradually decrease from 29.7 °C to 28.03 °C in the simulation, while the outlet temperatures decrease from 30.3 °C to 27.8 °C in the experiment. Here, the similarity is quite high between the simulation and the experiment results, with the maximum error of 1.9%. 

### 3.4. Pressure Distribution at R744 Side in Numerical Simulation

Figure 10 shows the pressure distribution on the R744 side for Case 2. The pressure profile result shows that the pressure value has an uneven decrease. However, the result is quite evenly distributed between the layers, and the pressure drop is uniform for the mini tube inside. When passing through the manifolds, R744 creates vortex zones that increase pressure loss at the manifolds. The pressure at the inlet of the unit is 72.83 bar and falls at the outlet with a value of 72.6 bar. The above results show that the pressure drop between the inlet and outlet of the unit is 0.23 bar.

### 3.5. The Effect of the Mass Flow Rate on the Pressure Drop

Figure 11 shows the relationship between the inlet pressure of the condenser and the mass flow rate of R744, both numerically and experimentally. The study included 30 steady states to figure out the pressure drop when R744 passes through the condenser while the outlet pressure is kept constant (green color point), and the inlet pressure is collected and verified with the experimental data. It is observed that the mass flow rates increase from 14.34 to 21.6 kg/h, the pressure drops are relatively stable at 0.4 bar (the inlet of 73 bar and the outlet of 72.6 bar) in the experiment; the pressure drops increase negligibly from 0.23 bar to 0.25 bar in the simulation. However, when increasing the mass flow rate from 25.08 to 46.08 kg/h, the pressure drop from the experiment is relatively stable at 0.5 bar while it changes from 0.3 bar to 0.47 bar in the simulation. This phenomenon is caused by the fact that the flowmeter in the experiment exhibits a high error at low flow values. When the flow increases, the accuracy of the measuring device is improved. To sum up, the numerical simulation can predict the pressure drop with greater accuracy at high values of the mass flow rate.

Figure 12 shows a comparison of the pressure drop between Case 1 (5 layers) and Case 2 (8 layers) in the experiment. The mass flow rates increase from 14.9 to 44.9 kg/h, and the pressure drops increase from 0.86 bar to 1.07 bar for Case 1; the pressure drops increase from 0.4 bar to 0.5 bar for Case 2. The pressure drops of Case 1 are much higher than those obtained from Case 2. This phenomenon is caused by the fact that the flowing path of R744 being longer with more passes than in the case of eight layers, which are divided into two sets (only four passes). The local pressure drop and the friction pressure drop generated are higher, leading to an increase in the overall pressure drop. 

### 3.6. The Comparison of the Optimal Design between Two Case Studies

In a typical case, the thermodynamic parameters when operating at the same cooling capacity of 960 W are given in Table 8 for comparing the two case studies.

Based on the data obtained for Case 1 (five layers), it is observed that the pressure drop is 1.07 bar, which is quite high compared to the Case 2 (eight layers) pressure drop with a value of 0.4 bar; so, this causes an increase in the adiabatic compression. At the same time, the log mean temperature difference is 8.56 °C for Case 1 compared to 14.7 °C for Case 2, demonstrating the heat transfer efficiency. In addition, the efficiency of the system is expressed with a COP value of 4.34 in Case 2, and it is quite high when compared with Case 1 (only 2.72). Thus, the configuration of eight layers divided into two sets is more optimal in the distribution of R744 to ensure a low-pressure drop, increasing the efficiency factor of the system.

From the results above, the evaporative condensers using mini tubes that are flooded in the cooling water tank are suitable for the subcritical R744 air conditioning system. This method could decrease the inlet temperature of the throttle valve and increase system efficiency. 

## 4. Conclusions

Two evaporative condensers using mini tubes were simulated with the numerical method using the COMSOL Multiphysics 6.1 software. The numerical results were verified with the experimental method. The study has achieved the following results.

The temperature field and the phase change were simulated under specific operating conditions; the outlet R744 temperature of the condenser changed from 28.7 °C to 30.3 °C, combined with the operating pressure that changed from 72.6 bar to 68.5 bar. The numerical results showed the completely liquefied state of R744.

Within the study limits, the mass flow rate increases from 14.34 kg/h to 46.08 kg/h, and the pressure drop also increases from 0.23 bar to 0.47 bar for the simulation and from 0.4 bar to 0.5 bar for the experiment, respectively. A turbulent flow increases the local pressure drop at the manifold, which in turn affects the pressure drop throughout the evaporative condenser. 

For the design of condensers, it is essential to arrange the number of tube layers so as to achieve higher heat transfer efficiency while reducing the pressure drop. The pressure drop of the five-layer tube configuration is higher than that obtained from the eight-layer one (splitting into two sets).

The evaporative condensers using mini tubes that are flooded in the cooling water tank are suitable for the subcritical R744 air conditioning system. The method could decrease the inlet temperature of the throttle valve and increase system efficiency. 

The numerical results are in good agreement with those obtained from the experimental results, with the maximum percentage of errors being less than 5%. These results will make additional contributions to numerical simulation studies of evaporative condensers, especially with CO_2_.

In addition, further research will focus on exploring tubes with smaller diameters, such as microscale, tube numbers, water velocity, etc., in harsher climatic conditions using the numerical and experimental methods.

## Figures and Tables

**Figure 1 micromachines-14-01826-f001:**
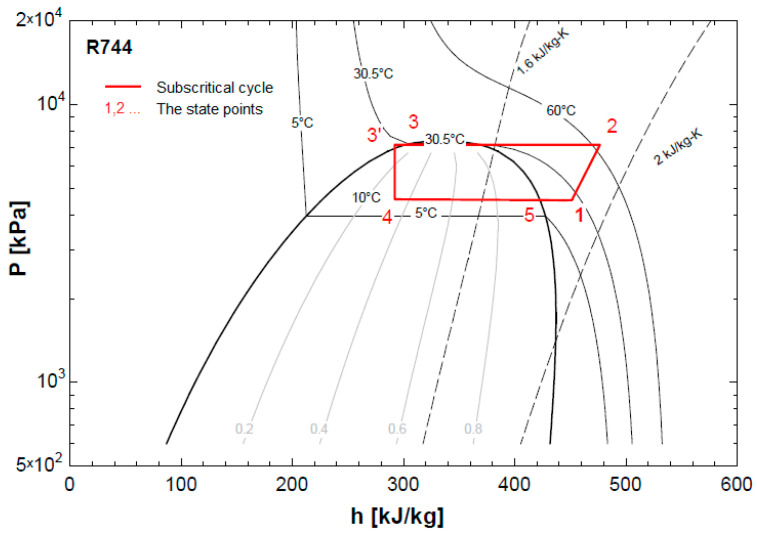
The P–h diagram of testing cycle.

**Figure 2 micromachines-14-01826-f002:**
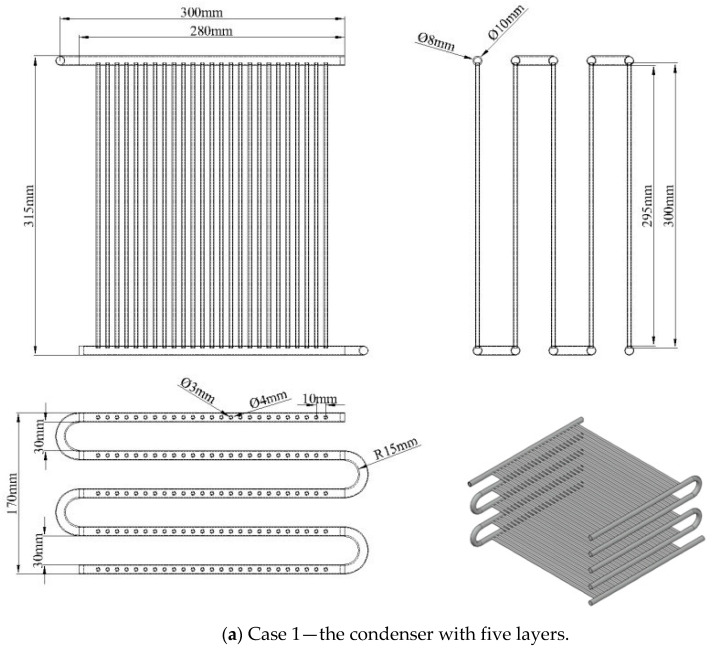
The design parameters of two condensers using mini tubes.

**Figure 3 micromachines-14-01826-f003:**
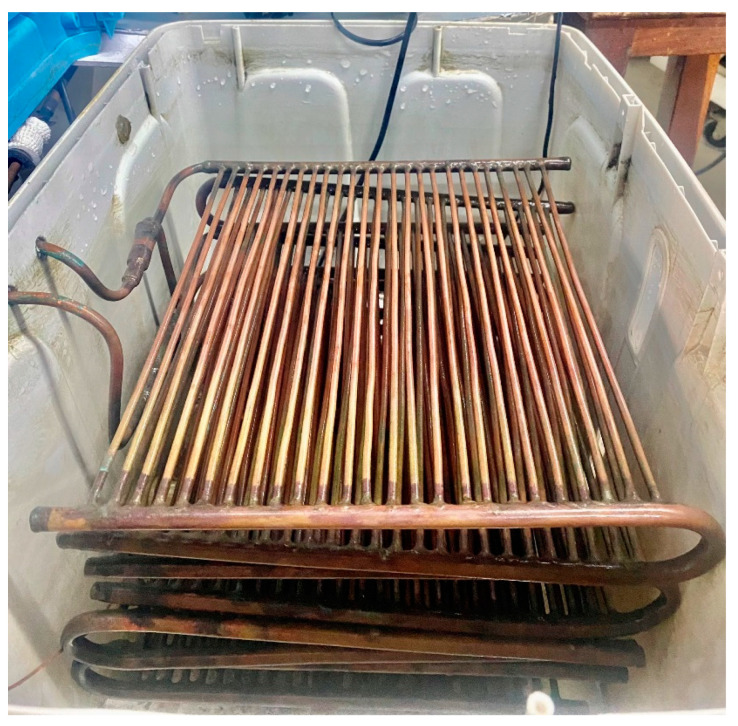
The actual photo of the condenser with eight layers using mini tubes.

**Figure 4 micromachines-14-01826-f004:**
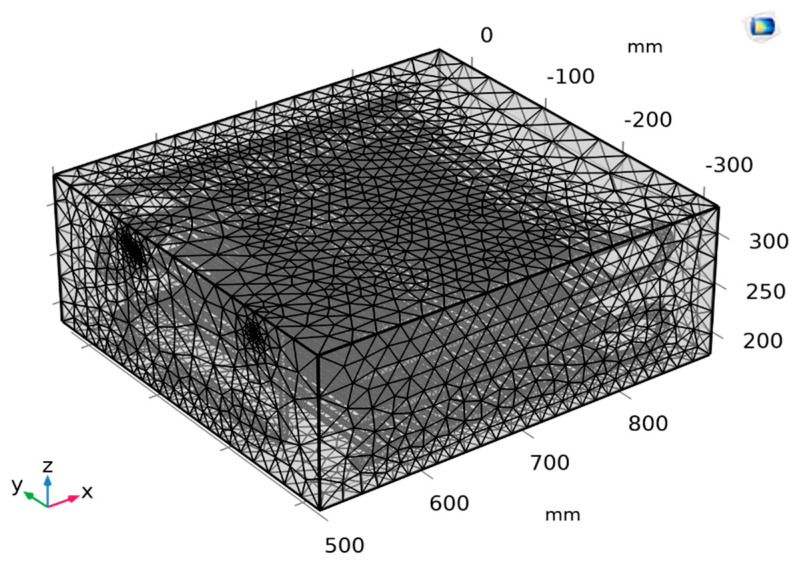
Meshing for the test sample.

**Figure 5 micromachines-14-01826-f005:**
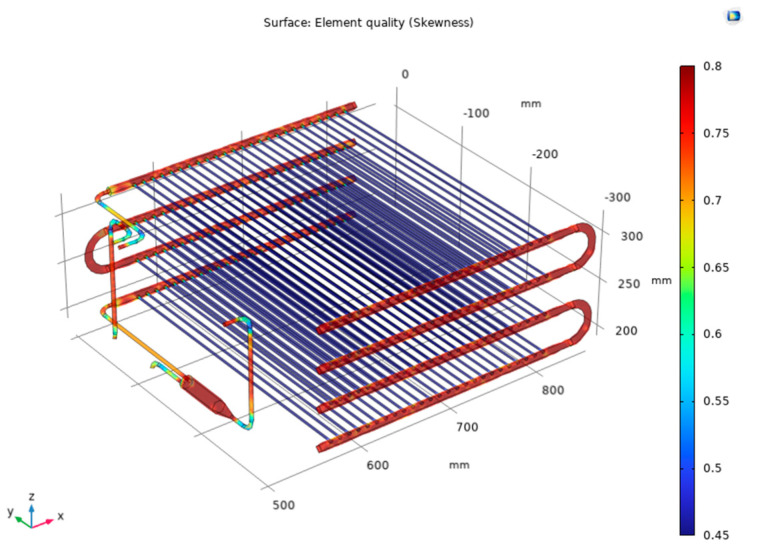
Skewness index for simulation model.

**Figure 6 micromachines-14-01826-f006:**
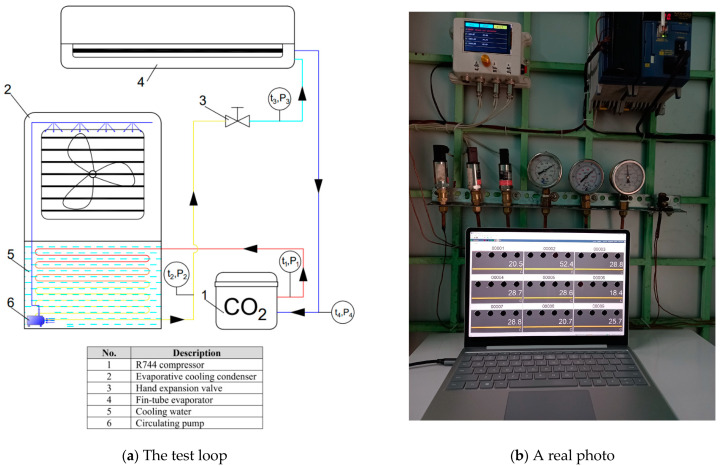
The diagram and a real photo of the test loop.

**Figure 7 micromachines-14-01826-f007:**
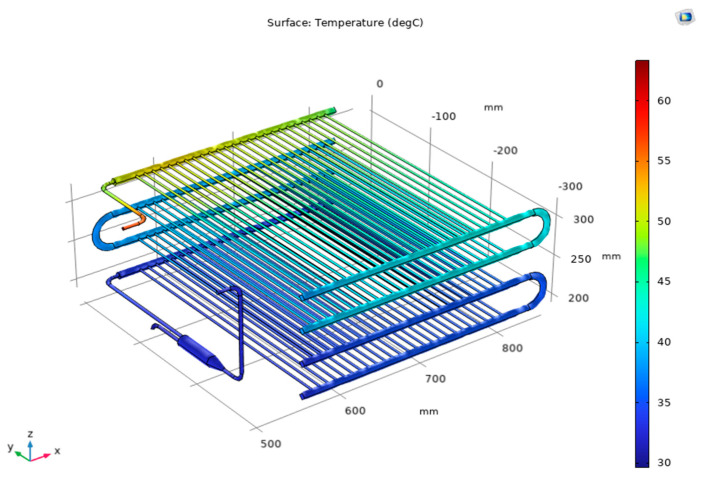
Temperature field at R744 side for Case 2.

**Figure 8 micromachines-14-01826-f008:**
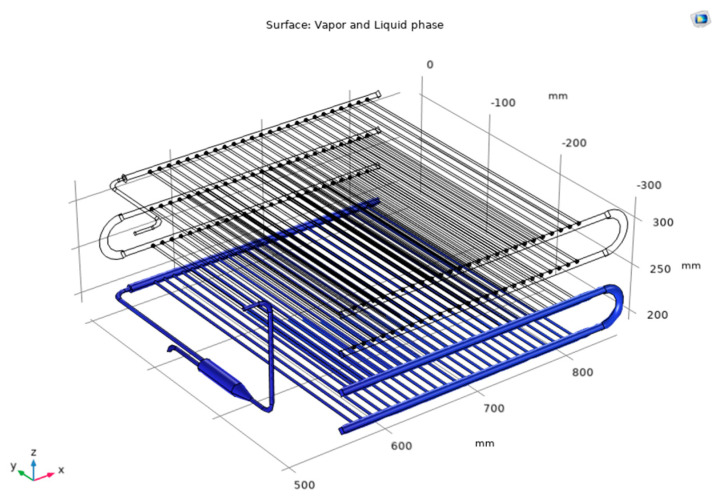
Phase indication at R744 side for Case 2.

**Figure 9 micromachines-14-01826-f009:**
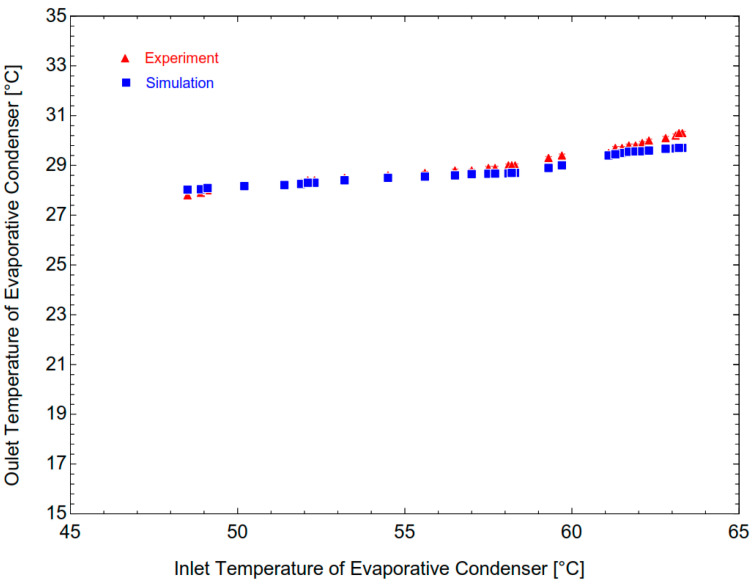
The inlet and outlet R744 temperatures of condenser from simulation and experiment.

**Figure 10 micromachines-14-01826-f010:**
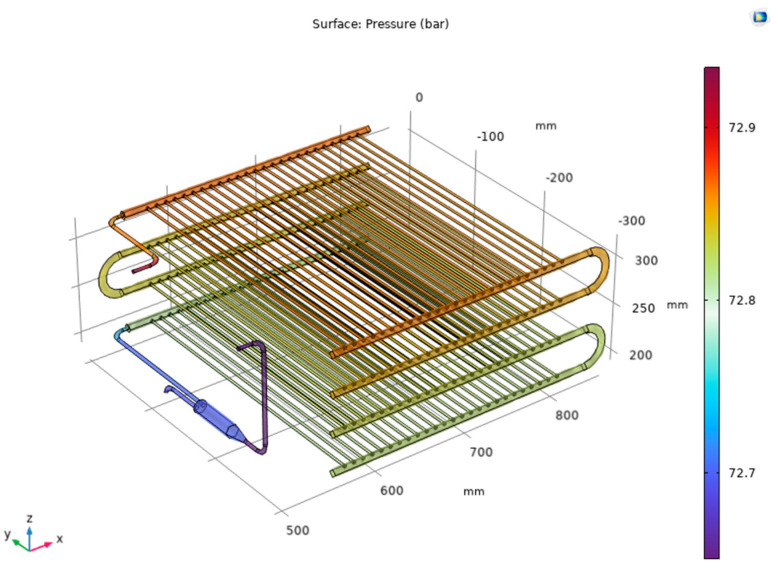
Pressure distribution on the R744 side.

**Figure 11 micromachines-14-01826-f011:**
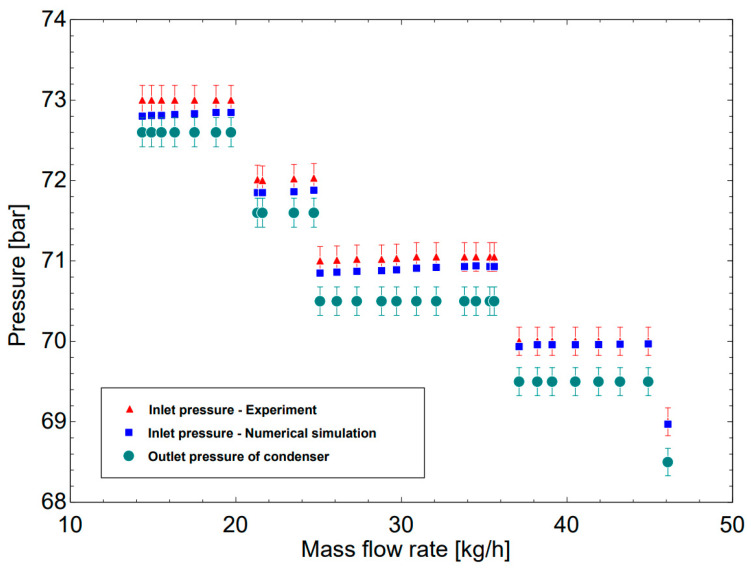
A comparison between experiment and numerical simulation on the inlet pressure with an increase in R744 mass flow rate.

**Figure 12 micromachines-14-01826-f012:**
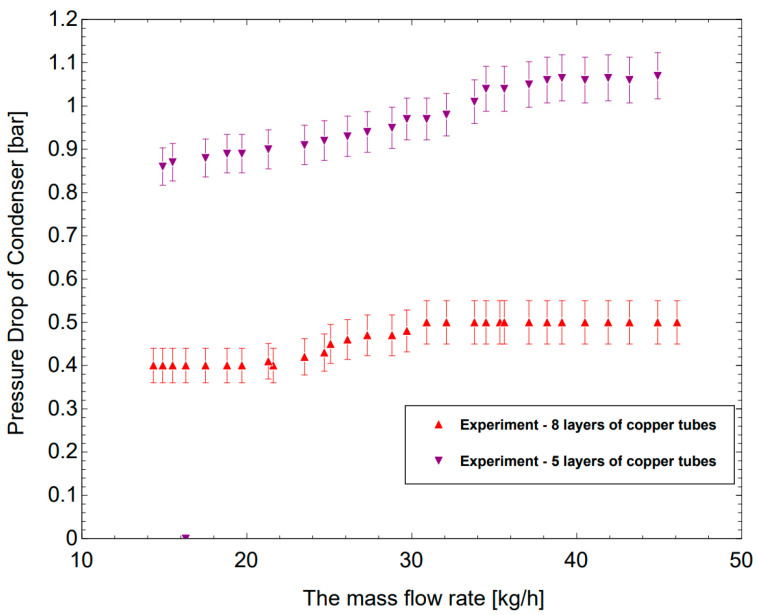
A pressure drop comparison between two cases with increasing of R744 mass flow rate.

**Table 1 micromachines-14-01826-t001:** Thermodynamic parameters at state points according to the initial design.

State Points	Temperature (°C)	Pressure (bar)	Enthalpy (kJ.kg)	Entropy (kJ.kg)	Volume (m^3^/kg)
1	22	45	444	1.89	0.0086
2	64	73	474	1.85	-
3	30.5	73	309	1.28	0.0016
3′	28.5	73	289	1.28	-
4	10	45	289	1.32	-
5	10	45	422	1.78	0.007

**Table 2 micromachines-14-01826-t002:** Design parameters for two configurations.

Parameters	Case 1 Five Layers	Case 2 Eight Layers
Overall heat transfer rate (W·m^−2^·K^−1^)	120.4	
Log mean temperature difference (°C)	17.6	11.07
Heat transfer area (m^2^)	0.47	0.75
Reynolds number range	1000–3000	
The number of pipes per layer	25	25
Overall dimension L × W × H (mm)	315 × 300 × 170	305 × 300 × 255
Number of passes	5	4
Number of sets	1	2

**Table 3 micromachines-14-01826-t003:** Grid independence verification.

Mesh Type Predefined	Grid Elements	Simulation
Coarse	232,519	28.62
Normal	692,399	28.70
Fine	1,184,533	28.71
Finer	7,634,022	28.81

**Table 4 micromachines-14-01826-t004:** Measuring devices and their accuracies.

Devices	Accuracy	Range	Manufacturers
Thermometer	±0.1 °C	−270–400 °C	Extech
Thermocouples–T type	±0.1 °C	0–100 °C	Omega-USA
Clamp meter	±2%FS	0–600 A	Hioki
Humidity meter	±3%FS	1.0–99.9%	Tenmars
Pressure sensor	0.5%FS	0–100 bars	Sensys
Turbine flow rate sensor	0.5%	400 to 5000 kg/s	DGT

**Table 5 micromachines-14-01826-t005:** Boundary conditions for the numerical simulation.

Input Parameters	Setting Value
Inlet temperature of R744 (°C)	63.3
Outlet pressure of R744 (bar)	72.6
Mass flow rate of R744 (g·s^−1^)	3.28
Average water temperature (°C)	26.4
Velocity of water (m/s)	0.003

**Table 6 micromachines-14-01826-t006:** Comparing between experiment and simulation of the condensers for Case 2.

Parameters	Unit	Experiment	Simulation	Error
Outlet R744 temperature	°C	30.3	29.7	1.9%
Outlet R744 pressure	bar	72.6	72.83	0.3%

**Table 7 micromachines-14-01826-t007:** Boundary conditions for the numerical simulation.

Input Parameters	Setting Range
Inlet of R744 (°C)	48.5–63.3
Outlet pressure of R744 (bar)	68.5–72.6
Mass flow rate of R744 (g·s^−1^)	3.28–6.89
Average water temperature (°C)	24.9–26.4
Velocity of water (m/s)	0.003

**Table 8 micromachines-14-01826-t008:** A comparison of thermodynamic parameters in two case studies.

Parameters	Case 1	Case 2
Log mean temperature difference (°C)	8.56	14.7
Pressure drop (bar)	1.07	0.4
Coefficienct of Performance	2.72	4.34
Adiabatic compression energy (W)	353	214

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
