# Peer review of "A Study on the Simulation and Experiment of Evaporative Condensers in an R744 Air Conditioning System"

_micromachines, 2023, doi:10.3390/mi14101826_

Round 1
Reviewer 1 Report
The authors presented a numerical and experimental study on the Evaporative Condensation in an R744 Air Conditioning System. The idea of the paper is interesting, but the scientific soundness is relatively low; the following points are to be addressed:
The main quantitative results are to be presented in the abstract.
The novelty of the paper is to be clearly stated.
The boundary conditions are to be expressed numerically.
The used turbulence model is to be justified.
The presented mesh seems to be coarse.
A grid sensitivity test is to be performed.
Line 154, write ‘’ Reynold’’ instead of ‘’ Renold’’
In the numerical model have you considered the tubes having thickness or it was neglected?
What is the considered range for Reynold number?
How is the phase change numerically treated? The presented equations don’t correspond to a multiphase flow; to be explained.
Information about the characteristics of the used computer and computation time are to be indicated.
An experimental uncertainty study is to be performed.
By examining fig 3, it seems that the tube layers are not equidistant (compared to the plotted design)
The scientific soundness is to be improved by adding physical interpretations to the discussion.
The paper is submitted to the special issue ‘’Heat Transfer and Fluid Flow in Microstructures’’ but the considered geometry is not in the size of micro; to be explained.
English level is low.
English level is low.
Author Response
Micromachines-2591435
Manuscript Title: A Study on the Simulation and Experiment of Evaporative Condensers in an R744 Air Conditioning System
Dear Editor and Reviewer #1:
Thank you for your comments and suggestions on the structure and content of our manuscript. We have revised the manuscript accordingly, and detailed responses to the comments and suggestions are listed below:
1) The main quantitative results are to be presented in the abstract.
>> We appreciate the comments provided by Reviewer #1. Following your comment, we have revised our abstract. It is noted that all revisions and additions to the original manuscript were marked in red.
2) The novelty of the paper is to be clearly stated.
>> Thanks for your deep comment. Following your comment, we have revised the manuscript especially in the Abstract, Results and Conclusion sections.
3) The boundary conditions are to be expressed numerically.
>> Per the review comments, we have revised to be clearer, which is indicated in Table 5 and Table 7.
4) The presented mesh seems to be coarse. A grid sensitivity test is to be performed.
>> Thanks for your deep comment. We have added Table 3 regarding grid verification.
5) Line 154, write ‘’ Reynold’’ instead of ‘’ Renold’’.
>> Many thanks for your kind comment, we have revised this error typing in Line 154 (Line 165 in the new revision).
6) In the numerical model have you considered the tubes having thickness or it was neglected?
>> In this study, the tube thickness is constant for both numerical and experimental methods. Thank you for your idea, the study on the effect of tube thickness will be carried out in the future.
7) What is the considered range for Reynold number?
>> The average Reynolds number inside the bank tube ranges from 1000 to 3000. We have added this information to the manuscript.
8) How is the phase change numerically treated? The presented equations don’t correspond to a multiphase flow; to be explained.
>> Thank you for your deep comments, we have added the phase change equation in Section 2.2.
9) Information about the characteristics of the used computer and computation time are to be indicated.
>> The computer for solving has a configuration as follow:
- CPU: Intel Core i7 6820HQ (2.7 GHz Turbo 3.6 GHz, 4 cores, 8 threads)
- Ram: 16GB DDR4 bus 2400 MHz
- VGA: Nvidia Quadro M3000M
The average computation time fluctuates from 30 minutes to 2 hours. We have added this information in Section 2.2.
10) An experimental uncertainty study is to be performed.
>> Following your deep comment, we have added the experimental uncertainty in Section 2.3.
11) By examining fig 3, it seems that the tube layers are not equidistant (compared to the plotted design)
>> Thanks for this comment. The purpose of the experimental model design process is to reduce the height of the condenser in the case of 8 layers. Due to a difference between theory and manufacturing technology on an experimental scale, there is a certain inclination, but the effect is negligible.
12) The scientific soundness is to be improved by adding physical interpretations to the discussion.
>> Thanks for this comment. We have revised by adding physical interpretations to the discussion throughout the paper. It is noted that all revisions and additions to the original manuscript were marked in red.
13) The paper is submitted to the special issue ‘’Heat Transfer and Fluid Flow in Microstructures’’ but the considered geometry is not in the size of micro; to be explained.
>> Thank you for your kind suggestion, micro size and mini size belong to the compact scale. In this study, we are using mini tubes in the evaporative condenser; hence, this situation can be acceptable.
14) English level is low.
>> Many thanks for your kind advice, after revising the entire technical comments, this manuscript was revised by a native English - speaking lecturer.
The revised paper (Second version) is resubmitted to your journal. Again, your efforts in reviewing the manuscript to make it more presentable and in publishing the paper are deeply appreciated.
Best Regards.
Reviewer 2 Report
Comments
In this article, Using R744, numerical simulations and experiments have been conducted on evaporative condensers in air conditioning systems. The topic of this research is interesting. I would suggest the publication of this article if the following minor issues are addressed.
- In the abstract, line9, “It indicated the capable of evaporative cooling….” Should be rephrased to be grammatically correct.
- In the introduction more recent works should be added. I suggest to add the following reference:
Nidhal Ben Khedher, Azfarizal Mukhtar, Ahmad Shah Hizam Md Yasir, Nima Khalilpoor, Loke Kok Foong, Binh Nguyen Le & Hasan Yildizhan (2023) Approximating heat loss in smart buildings through large scale experimental and computational intelligence solutions, Engineering Applications of Computational Fluid Mechanics, 17:1, DOI: 10.1080/19942060.2023.2226725
- Table2, the overall heat transfer coefficient is it the same for the two cases (5 and 8 layers)? Please justify this and how this coefficient is evaluated or estimated.
- Section 2.2, in the numerical model the boundary conditions are missing and should be added and detailed properly.
- In table 6, the pressure drop is higher for case 1 compared to case 2 which is illogical since the number of layers is greater for condenser 2.
- Section 3.6, the validation should be placed immediately after section 3.1
- Table 4 caption, there is a missing s in the caption precisely in boundary conditions
There are many grammatical errors and typos. Please revise the whole text.
Author Response
Micromachines-2591435
Manuscript Title: A Study on the Simulation and Experiment of Evaporative Condensers in an R744 Air Conditioning System
Dear Editor and Reviewer #2:
Thank you for your comments and suggestions on the structure and content of our manuscript. We have revised the manuscript accordingly, and detailed responses to the comments and suggestions are listed below:
1) In the abstract, line9, “It indicated the capable of evaporative cooling….” Should be rephrased to be grammatically correct.
>> We appreciate the comments provided by Reviewer #2. We rephrased to make this sentence clearly. It is noted that all revisions and additions to the original manuscript were marked in red.
2) In the introduction more recent works should be added. I suggest to add the following reference:
Nidhal Ben Khedher, Azfarizal Mukhtar, Ahmad Shah Hizam Md Yasir, Nima Khalilpoor, Loke Kok Foong, Binh Nguyen Le & Hasan Yildizhan (2023) Approximating heat loss in smart buildings through large scale experimental and computational intelligence solutions, Engineering Applications of Computational Fluid Mechanics, 17:1, DOI: 10.1080/19942060.2023.2226725
>> Thanks for your kind suggestions, we have added several recent studies in our Introduction section, especially with the reference [10].
3) Table2, the overall heat transfer coefficient is it the same for the two cases (5 and 8 layers)? Please justify this and how this coefficient is evaluated or estimated.
>> In this study, the overall heat transfer coefficient is constant, the heat transfer areas between 2 cases are different, so the temperature difference is also different. The heat transfer coefficient of the R744 side is very high while the water side is quite low; hence, the overall heat transfer coefficient almost depends on the heat transfer coefficient of the water side. We estimate based on theoretical calculations of the R744 condensation process with an approximate value of 120.4 W/m2.K (mentioned in Table 2).
4) Section 2.2, in the numerical model the boundary conditions are missing and should be added and detailed properly.
>> Thanks for your deep comment. We have revised it to be clearer, which is indicated in Table 5 and Table 7.
5) In table 6, the pressure drop is higher for case 1 compared to case 2 which is illogical since the number of layers is greater for condenser 2
>> Thanks for your deep comments. The case study of 8 layers is divided into 2 sets with 4-layer per set compared to 5-layer. The aim of this work is to verify the pressure drop in simulation and experiment and evaluate the heat transfer efficiency for the two configurations.
6) Section 3.6, the validation should be placed immediately after section 3.1
>> Thanks for your kind suggestion, we have revised following your comment.
7) Table 4 caption, there is a missing s in the caption precisely in boundary conditions
>> Thanks for your kind comment, we have revised it.
8) There are many grammatical errors and typos. Please revise the whole text.
>> Many thanks for your kind advice, after revising the entire technical comments, this manuscript was revised by a native English - speaking lecturer.
The revised paper (Second version) is resubmitted to your journal. Again, your efforts in reviewing the manuscript to make it more presentable and in publishing the paper are deeply appreciated.
Best Regards.

Reviewer 3 Report
The authors presented results of the numerical simulation and experimental investigation of evaporative condensers using R744 for the air conditioning system. Two design configurations using mini tubes were analyzed and compared. The topic of this manuscript is relevant to energy and environmental issues. The text of the article is well structured and well presented. Some comments are suggested to improve the article:
The abstract describes the subject of the paper and the results of the study. The problem should be described a little more fully and attention should be paid to the research methods used.
The abbreviation IHX in line 53 must be explained.
Authors should clearly state the purpose of the study at the end of the Introduction.
It is not clear why the dot is placed after the quantity at the bottom in the equations; it should be a multiplication sign (Section 2.2).
All quantities in Section 2.2 must be explained.
Revise typing of turbulence model: Reynolds or Renold (line 154).
How long does each experiment take?
Figure 6 has two pictures. Only a) and b) should be used.
As can be seen in Fig. 7. The discrepancy in the results increased at higher inlet temperatures of the evaporative condenser. What do you think might have caused this?
Revise typing: line 312.
Conclusions should be complemented by future plans.
Typing errors should be revised.
Author Response
Micromachines-2591435
Manuscript Title: A Study on the Simulation and Experiment of Evaporative Condensers in an R744 Air Conditioning System
Dear Editor and Reviewer #3:
Thank you for your comments and suggestions on the structure and content of our manuscript. We have revised the manuscript accordingly, and detailed responses to the comments and suggestions are listed below:
1) The abstract describes the subject of the paper and the results of the study. The problem should be described a little more fully and attention should be paid to the research methods used.
>> Thanks for your kind comments, we have revised the abstract as well as the whole paper following your comments. It is noted that all revisions and additions to the original manuscript were marked in red.
2) The abbreviation IHX in line 53 must be explained.
>> I apologize for this inconvenience. We have explained IHX in line 53.
3) Authors should clearly state the purpose of the study at the end of the Introduction.
>> Thanks for your deep comment, we have revised the manuscript espescially in Introduction sections.
4) It is not clear why the dot is placed after the quantity at the bottom in the equations; it should be a multiplication sign (Section 2.2).
>> We have revised following your comment. Many thanks for your kind advice.
5) All quantities in Section 2.2 must be explained.
>> We have checked and explained following your comments.
6) Revise typing of turbulence model: Reynolds or Renold (line 154).
>> Many thanks for your kind comment, we have revised this error typing in Line 154. (Line 165 in the new revision)
7) How long does each experiment take?
>> Thanks for this comment. The test loop operated for 20 minutes in order to have the system reach its steady state. Then, the experimental data (temperature, pressure, flow rate, etc.) were recorded for 10 minutes. The values for the system as a whole were calculated by the average values of all temperatures being recorded.
8) Figure 6 has two pictures. Only a) and b) should be used.
>> Thanks for this comment. We have checked and separated by two figures (Figure 6 and Figure 7).
9) As can be seen in Fig. 7. The discrepancy in the results increased at higher inlet temperatures of the evaporative condenser. What do you think might have caused this?
>> Many thanks for your kind comments. This phenomenon occurs due to the measurement device uncertainty, which is still within acceptable limits; in fact, the temperature difference is less than 1oC. We adjusted the scale in Figure 8 (the initial number is 7) to make the results more consistent.
10) Revise typing: line 312.
>> Thanks for this comment. We have revised “according to” instead of “accordingly to”.
11) Conclusions should be complemented by future plans.
>> Thanks for your kind advice. We have revised following your comments in the Conclusion section.
12) Typing errors should be revised.
>> Many thanks for your kind advice, after revising the entire technical comments, this manuscript was revised by a native English - speaking lecturer.
The revised paper (Second version) is resubmitted to your journal. Again, your efforts in reviewing the manuscript to make it more presentable and in publishing the paper are deeply appreciated.
Best Regards.

Round 2
Reviewer 1 Report
After revision, the paper can be accepted for publication